# Quantitative characterization of extracellular vesicle uptake and content delivery within mammalian cells

Emeline Bonsergent [1,2], Eleonora Grisard[1], Julian Buchrieser [3], Olivier Schwartz[3], Clotilde Théry [1] & Grégory Lavieu [1,2 ✉]

Extracellular vesicles (EVs), including exosomes, are thought to mediate intercellular communication through the transfer of cargoes from donor to acceptor cells. Occurrence of EV-content delivery within acceptor cells has not been unambiguously demonstrated, let alone quantified, and remains debated. Here, we developed a cell-based assay in which EVs containing luciferase- or fluorescent-protein tagged cytosolic cargoes are loaded on unlabeled acceptor cells. Results from dose-responses, kinetics, and temperature-block experiments suggest that EV uptake is a low yield process (~1% spontaneous rate at 1 h). Further characterization of this limited EV uptake, through fractionation of membranes and cytosol, revealed cytosolic release (~30% of the uptaken EVs) in acceptor cells. This release is inhibited by bafilomycin A1 and overexpression of IFITM proteins, which prevent virus entry and fusion. Our results show that EV content release requires endosomal acidification and suggest the involvement of membrane fusion.

[1] Institut Curie, PSL Research University, INSERM U932, Paris, France. [2] Université de Paris, INSERM, CNRS UMR 7057, Paris, France. [3] Institut Pasteur, Virus and Immunity Unit, Department of Virology, CNRS, UMR 3569, Paris, France. ✉email: gregory.lavieu@inserm.fr

Extracellular vesicles, which include exosomes and micro-vesicles, contain cargoes such as nucleic acids, proteins and lipids[1]. As many other vectors of communication, EVs mediate cell–cell communication through activation of receptors located at the surface of acceptor cells[2]. In addition, EVs have been proposed to transfer membrane encapsulated cargoes from donor to acceptor cells[3–9], and impact the phenotype of the latter. EVs released by tumor cells are often described as instrumental in affecting tumor surrounding cells to favor their own growth and dissemination[10,11].

EVs are internalized by many cell types and through multiple pathways[1,12]. Endo/lysosomal targeting is often a pre-requisite for the content delivery of viruses, that in several cases is controlled by the acidic endo/lysosomal pH that triggers fusion between virus and endo/lysosomal membranes[13]. Our previous study that used a cell-free system[14], and others[15], suggest that EV-content delivery relies on similar pH-dependent mechanism. However, the true occurrence of EV-content release within acceptor cells remains debated, and has never been rigorously demonstrated, nor quantified. Until now, only EV uptake has been quantitatively assessed[16]. In addition, there is no consensus regarding the mode of EV uptake, which could be receptor-dependent or not. For instance, several cell surface molecules have been proposed to play a role in EV capture[17–21], but none of them seems required and sufficient. In addition, fate of EV cargoes within the acceptor cells is not characterized. EV cargoes could be released in the cytosol, degraded within lysosome, or re-secreted in the extracellular media through recycling within newly formed EV.

Here, we combined well-established quantitative classical biochemistry with qualitative and quantitative subcellular confocal imaging, to follow the fate of a generic EV cytosolic cargo and measure the actual efficacy of EV capture and content transfer into acceptor cells. Our results suggest that EV-content release requires endosomal acidification and membrane fusion.

## Results

**Validation of NLuc-Hsp70 as an EV-encapsulated cargo.** First, we engineered donor Hela cells stably expressing NanoLuc luciferase-tagged Hsp70 (hereafter named NLuc-Hsp70). We focused on Hsp70, an established generic EV marker[22–24], thus a good candidate for cytosolic release assessment, in contrast with most of the EV markers that are membrane-associated. Importantly, we successfully validated and used this EV cargo in our recent cell-free reconstitution study[14]. We used NanoLuc luciferase, a recently engineered luciferase that has superior signal-to-noise ratio, to ensure high sensitivity detection of our assay[25]. In-gel luciferase activity from lysates of cells that stably expressed NLuc-Hsp70, revealed that the chimeric proteins migrated, as expected, as a unique band corresponding to a >75 kDa protein (Fig. 1a). The absence of detectable partial degradation of NLuc-Hsp70 validated the relevance to monitor NLuc-Hsp70 behavior/fate exclusively through its enzymatic/NLuc activity.

We then isolated EVs released by NLuc-Hsp70 positive donor cells through sequential ultracentrifugation, and compared the luciferase activity between equal protein amount from cell lysate of donor cells and isolated EVs emanating from those cells (Fig. 1b). Enzymatic activities were similar, showing lack of cargo's enrichment within EVs. This is consistent with our previous results[14] and suggests bulk loading of Hsp70 within EVs. As expected, the EV fraction was positive and enriched for classical EV markers such as CD63, CD9, positive for Alix, and negative for Calnexin, an endoplasmic reticulum resident protein generally absent from EVs (Fig. 1c). In addition, and accordingly to the standards for studies of EVs[26], isolated vesicles were also qualitatively validated by electron-microscopy (Supplementary Fig. 1).

Throughout this study, we used the NLuc-CD63 chimeric protein as a control, alongside NLuc-Hsp70. To validate this control, we first performed in-gel luciferase activity on cells transiently expressing NLuc-CD63, which satisfyingly showed lack of significant NLuc-CD63 degradation (Fig. 1a). We also measured luciferase activity in EV and cell lysate and observed on average an enrichment of NLuc-CD63 in the EV fraction (Fig. 1b), consistent with the known enrichment of endogenous CD63.

We then investigated if NLuc-Hsp70 was truly inside EVs, and if the integrity of the EV was not compromised by the isolation procedure. We used a well-established proteinase protection assay, in which cargo contained in a vesicle can only be digested by a proteinase present in the buffer if membranes are disrupted by detergents[14,27,28]. As a positive control, we used NLuc-CD63, in which the luciferase enzyme was fused to the cytoplasmic N-terminal extremity of the tetraspanin CD63 (inside the vesicle). Both chimeric proteins behave similarly (Fig. 1d): in absence of detergent, roughly 80% of the NLuc activity was recovered for both cargoes ($75 \pm 5\%$ and $80 \pm 11\%$ for NLuc-Hsp70 and NLuc-CD63, respectively). In presence of detergent, NLuc activity dropped to <20% for both cargoes, suggesting that the luciferase was indeed inside the vesicles. This again is consistent with our previous study that used GFP-tagged Hsp70[14].

One possibility is that the NLuc-Hsp70 fraction that is sensitive to proteinase K is not associated with vesicles. To rule out this possibility, we further separated EVs from co-isolated non-vesicular components through floatation into sucrose[2,29] (Supplementary Fig. 2A). Briefly, $100,000 \times g$ pellets emanating from conditioned media of NLuc-Hsp70- or NLuc-CD63-expressing cells were resuspended in 60% sucrose buffer and overlaid with 2 layers of lower concentrations of sucrose buffer (30%, 0%) followed by overnight centrifugation. Three fractions were collected (top, middle, and bottom, respectively), diluted, subjected to $100,000 \times g$ centrifugation, and pellets were resuspended in PBS. More than 80% of the NLuc activity was found in the middle fraction, where EVs are expected to float, for both NLuc-Hsp70 and NLuc-CD63 cargoes (Supplementary Fig. 2B). Endogenous Alix was also more abundant within the middle fraction (Supplementary Fig. 2C). Middle-fraction EV were then subjected to the protease protection assay (Supplementary Fig. 2D). Consistently with the aforementioned results, in absence of detergent >80% of the NLuc activity was recovered for both cargoes ($86 \pm 4\%$ and $86 \pm 7\%$ for NLuc-Hsp70 and NLuc-CD63, respectively). NLuc activity dropped to <15% in the presence of detergent. Endogenous Alix, tested by immunoblot, showed similar behavior (Supplementary Fig. 2E). Our simplest explanation is that this modest but yet measurable "unexpected" degradation of EV cargo reflects partial damage of EV membranes induced by the isolation procedure.

We concluded that NLuc-Hsp70 is an appropriate cargo to monitor EV uptake and content delivery within acceptor cells.

**EV-uptake characterization.** We then loaded NLuc-Hsp70-positive EVs on unlabeled acceptor HeLa cells, to assess EV-mediated homotypic cell–cell transport. First, we performed dose–response and kinetic experiments. Luciferase activity recovered in EV-treated acceptor cells increased proportionally to the dose of donor EVs (Fig. 2a). Importantly, saturation could not be reached even at high doses of EVs (100 μg/ml of proteins and higher). Kinetics showed that luciferase activity in acceptor cells increased over time, with a 1% spontaneous rate at 1 h (Fig. 2b).

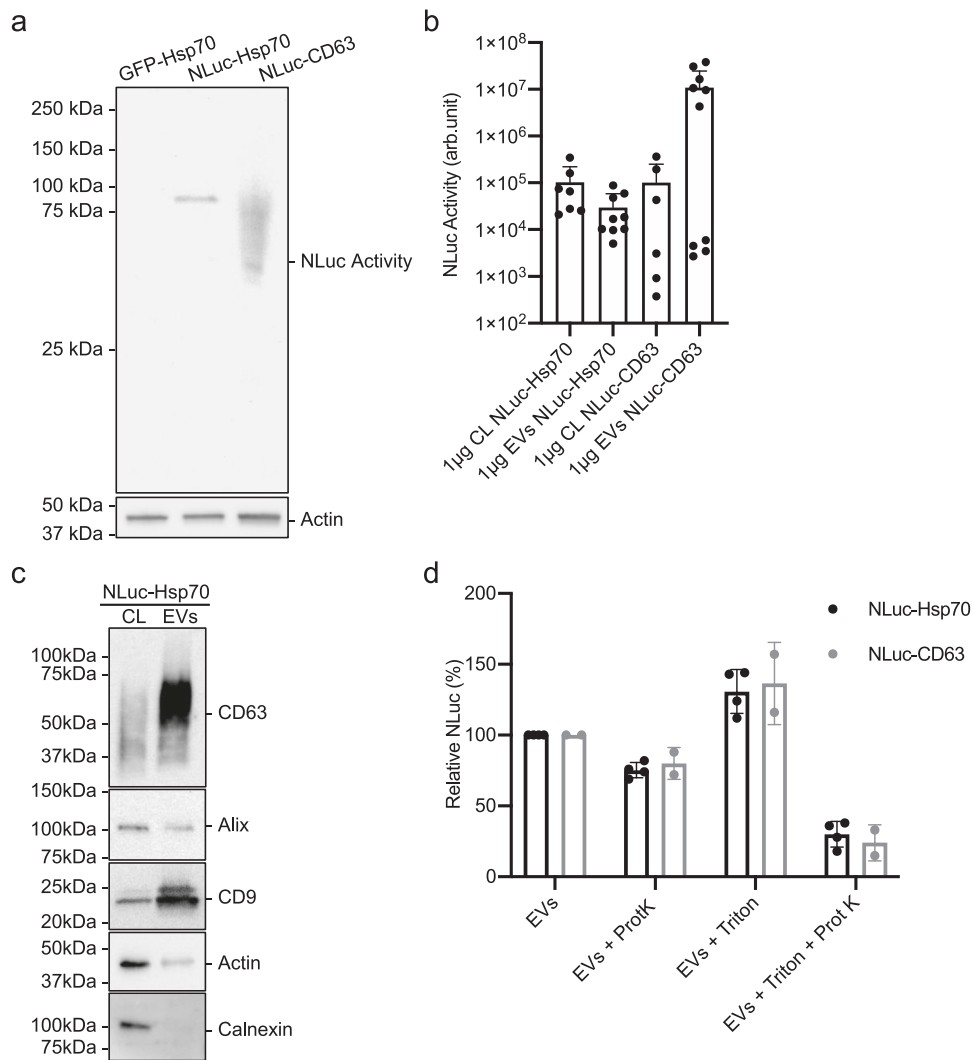

**Fig. 1 NLuc-Hsp70 and NLuc-CD63 EV characterization. a** In-gel detection of NLuc-Hsp70 and NLuc-CD63 activity. Equal protein amount of cell lysates from HeLa GFP-Hsp70 (negative control), stable HeLa NLuc-Hsp70, and transient HeLa NLuc-CD63 (upper gel) were loaded. As a control, actin was tested by immunoblot on the same samples (lower gel). This blot is representative of 2 independent experiments. **b** NLuc activity measurement in 1 μg of cell lysate (CL) or EVs from stable HeLa NLuc-Hsp70 or transient HeLa NLuc-CD63. Each dot is an independent replicate and represents the mean of 2 or 3 technical replicates. From left to right, $n = 7, 9, 6, 10$. Error bars represent standard deviations. **c** Immunoblots of cell lysate (CL) and EVs from stable HeLa NLuc-Hsp70. Equal amounts of protein were loaded to analyze CD9, CD63, Actin, Alix, and calnexin. This blot is representative of three independent experiments. **d** Protease protection assay on NLuc-Hsp70 (black) or NLuc-CD63 (gray) EVs, treated or not with proteinase K and/or detergent. Non-treated EVs were set to 100%. Each dot is an independent replicate and represents the mean of 3 technical replicates, $n = 4$ for NLuc-Hsp70, $n = 2$ for NLuc-CD63. Error bars represent standard deviations.

Importantly, we confirmed those results with EVs isolated through floatation assay (Supplementary Fig. 2F). We measured 1.2 ± 0.2% uptake for middle-fraction NLuc-Hsp70-EVs, whereas soluble recombinant NLuc was unable to penetrate the acceptor cells (0.03 ± 0.01% uptake). These additional control experiments demonstrate that we are indeed following the fate of EV-containing cargoes.

We then used 4 °C temperature block, known to inhibit energy-dependent endocytosis without affecting protein–protein interaction often required for cell surface docking. We reasoned that if EVs were captured at the surface of acceptor cells by a dedicated receptor, NLuc should be detectable even at such a low temperature. The luciferase activity that is normally associated with acceptor cells after EV treatment was virtually absent (0.4 ± 0.2%) in cells treated at 4 °C for up to 2 h, suggesting very poor, or lack of specific EV binding at the surface of acceptor cells (Fig. 2c).

We then used confocal imaging to monitor the fate of uptaken EVs and performed colocalization experiments between GFP-tagged Hsp70 emanating from donor EVs and endogenous markers for early endosomes (Rab5) and lysosomes (Lamp1) (Fig. 2d). We observed colocalization of GFP with both markers (Fig. 2e, f).

We concluded that EVs are indeed internalized in endo/lysosomal compartments.

**EV-content release characterization**. We reasoned that if EV-content release occurs, luciferase activity of NLuc-Hsp70 should be measurable within the cytosolic fraction, whereas if EV content remains trapped or degraded inside endo-lysosomal compartments, cytosolic fraction should be negative for luciferase activity (Fig. 3a). To separate cytosolic from membrane fractions we used a detergent-free cell fractionation (Fig. 3b).

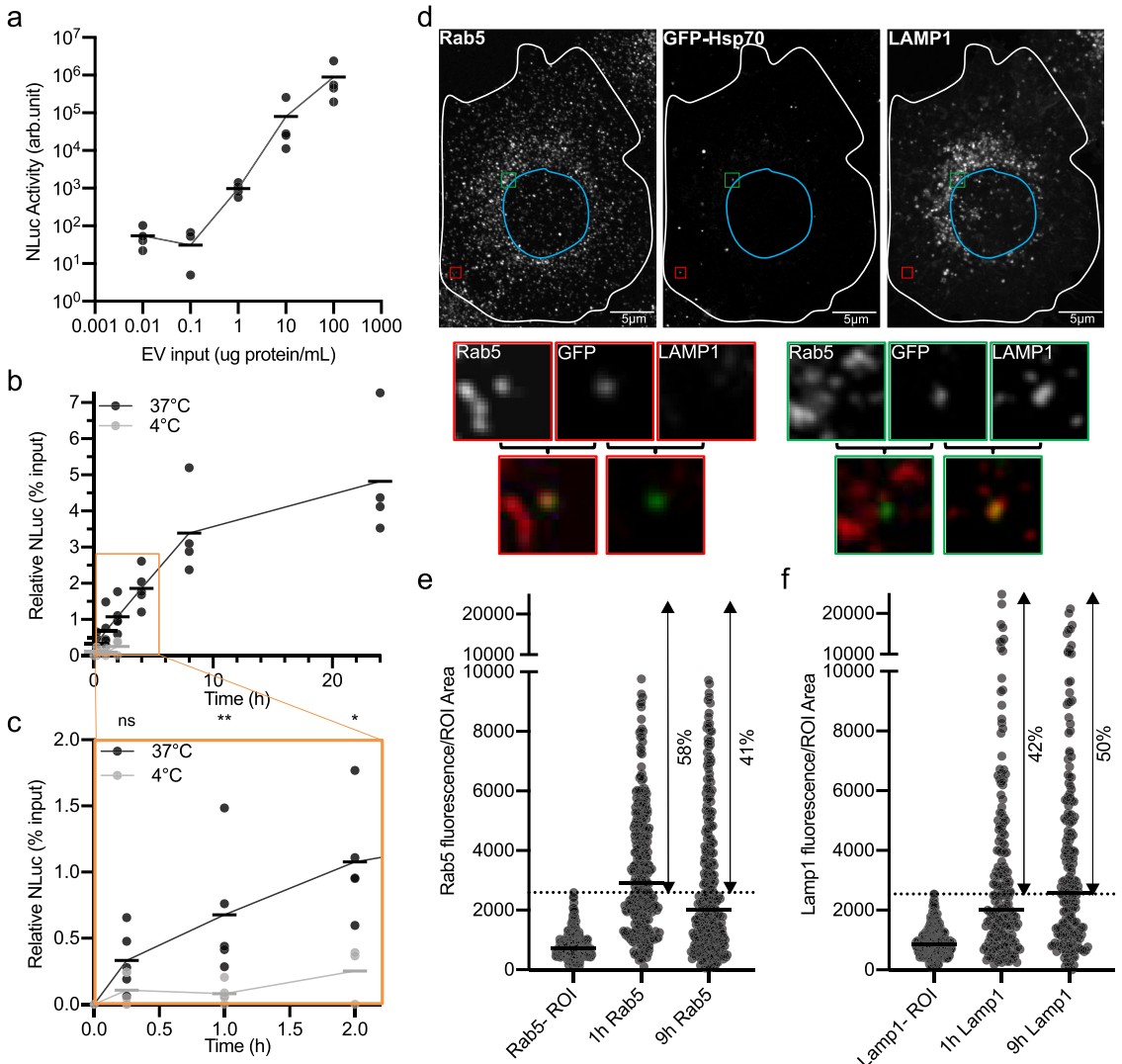

**Fig. 2 NLuc-Hsp70 EV uptake by HeLa cells. a** Dose–response study. Unlabeled acceptor HeLa WT cells were incubated for 2 h. with different amounts of isolated NLuc-Hsp70 EVs, luciferase activity was assessed at 2 h. One dot is an independent replicate and represents the mean of 2 technical replicates, $n$ = 4. **b** Kinetics study. Unlabeled acceptor HeLa WT cells were incubated with isolated NLuc-Hsp70 EVs (1 μg/ml) for different incubation times at 37 °C (black) or 4 °C (gray). Measured EV input (equivalent to the EV dose loaded on cells) was set to 100% to normalize the NLuc activity at each timepoint. Each dot is an independent replicate and represents the mean of 2 technical replicates, $n$ = 4. **c** Temperature-dependency study. Orange square, zoom on the 0–2 h timepoints on graph (**b**), to highlight 4 °C kinetics. Each dot is an independent replicate and represents the mean of 2 technical replicates, $n$ = 4. **d** Confocal micrographs showing GFP-Hsp70 EV-content colocalization with either endosomes (Rab5+, red square) or lysosomes (LAMP1+, green square) compartments. Unlabeled acceptor HeLa WT were incubated with GFP-Hsp70 EV for 1 h, then processed for immunofluorescence against Rab5 and LAMP1 prior to being imaged by confocal microscopy. Micrographs are representative of three independent experiments. **e** Quantification of GFP-Hsp70 EV-content colocalization with endosomal and lysosomal compartments. Rab5 fluorescence was measured for each GFP+ dot or negative ROI. Each dot is a GFP+ compartment or Rab5 negative ROI of the same size, $n$ = 295, 315, 291 (for each column, left to right order). Dashed line represents the maximum fluorescence of Rab5 negative ROIs. **f** Quantification of GFP-Hsp70 EV-content colocalization with lysosomal compartments. Lamp1 fluorescence was measured for each GFP+ dot or negative ROI. Each dot is a GFP+ compartment or Lamp1 negative ROI of the same size, $n$ = 193, 196, 195 (for each column, left to right order). Dashed line represents the maximum fluorescence of Lamp1 negative ROIs.

We first validated our cell fractionation protocol. Briefly, unlabeled HeLa cells were mechanically disrupted and submitted to sequential centrifugation steps to first remove undamaged cells and large debris (350 g) and then separate cytosolic from membrane fractions (100,000 × g). Supernatant (cytosol) and pellet (membrane) fractions were tested by western blotting for endogenous Hsp70 as a cytosolic marker, CD63 and Calnexin as membrane markers (Fig. 3c). Satisfyingly, >90% of the Hsp70 signal was recovered in the cytosolic fraction, which contains <10% of the CD63 and calnexin, validating our ability to separate membranes and cytosol (Fig. 3c, d, and f). Finally, when samples were exposed to detergent (Triton X-100), all markers

were recovered almost exclusively in the cytosolic fraction, as expected (Fig. 3c, d, and f). Because our delivery assay depends on measurement of luciferase activity, we performed the very same tests in cells expressing NLuc-Hsp70 or NLuc-CD63. Consistent with previous results, cytosolic fraction contained 90% of luciferase activity in NLuc-Hsp70-expressing cells and <10% in NLuc-CD63-expressing cells (Fig. 3e). This validated our method.

We then loaded NLuc-Hsp70- or NLuc-CD63-positive EVs on unlabeled acceptor cells for 1–4 h prior to performing fractionation and measuring luciferase activity in membrane or cytosolic fractions. For NLuc-Hsp70, $27 \pm 7\%$ of the cell-associated luciferase activity that corresponds to the internalized EVs was

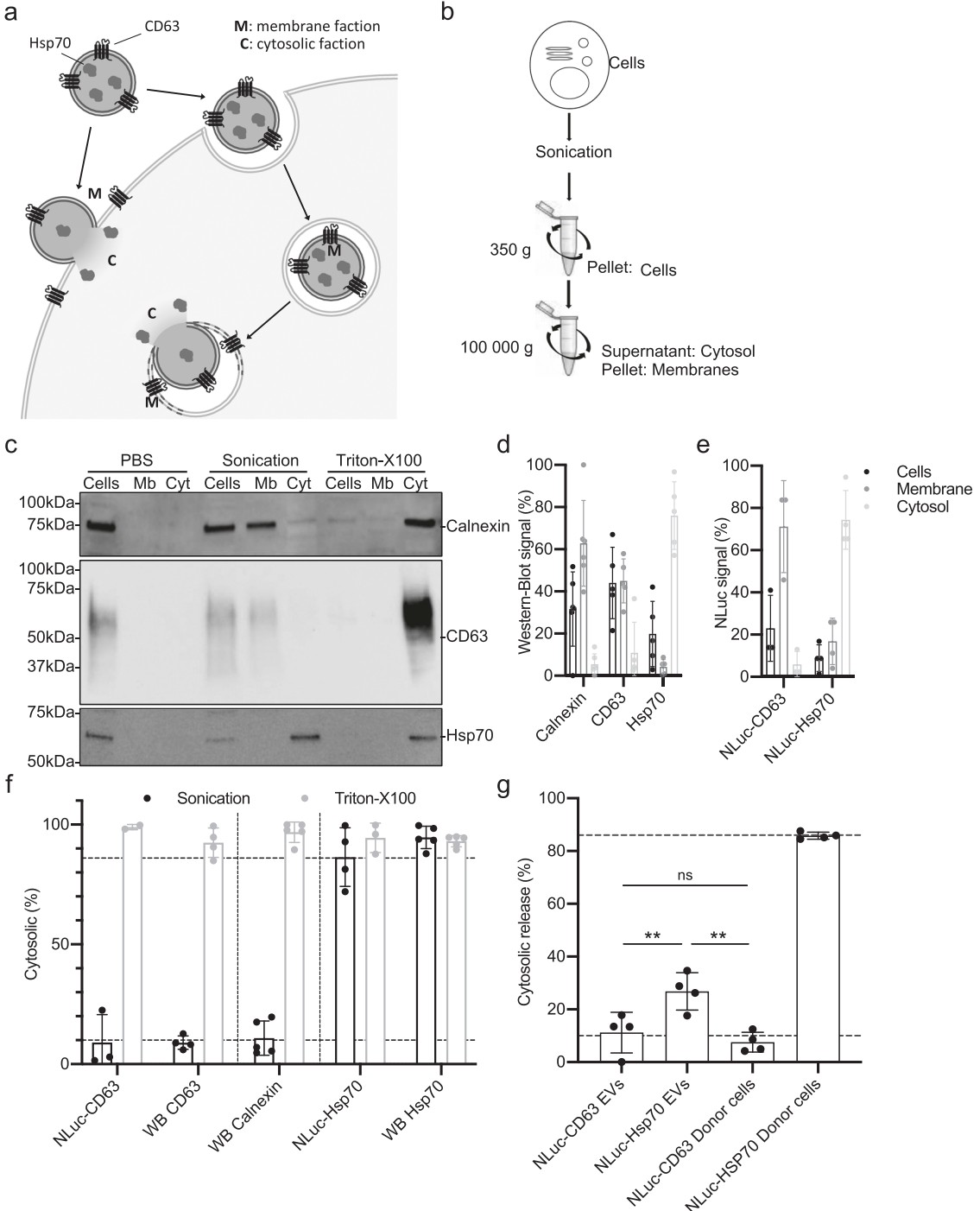

recovered in the cytosolic fraction (Fig. 3g). Cytosolic release of NLuc-CD63 originally emanating from donor EVs was 11 ± 8% (Fig. 3g), consistent with the behavior of NLuc-CD63 within the donor cells (Fig. 3e) that corresponds to background signal). This key data demonstrated that roughly 20–30% of the internalized EVs are capable of releasing their soluble content within the cytosol of acceptor cells (Fig. 3g).

Importantly, treatment of acceptor cells with bafilomycin A1, which inhibits endosome acidification, decreased cytosolic release of NLuc-Hsp70 while general EV uptake was unchanged (Fig. 4a and b). Confocal microscopy that enables the tracking of donor GFP-Hsp70 EVs, revealed an increased number of GFP-Hsp70 foci within the cell (Fig. 4c and d); those GFP foci co-localize with

endosomes (Supplementary Fig. 3A) consistent with EV confinement within neutralized endo/lysosomes.

One possibility is that internalized and partially digested EVs could damage endosomal membrane to trigger EV-content release. To test this hypothesis, we used antibody against endogenous galectin-3, a cytosolic protein that decorates the luminal face of endosomes that have lost their membrane integrity[30]. As a positive control we used LLOME, known to disrupt endosomal membrane[2]. As expected, LLOME induced galectin-3 foci, whereas internalized EVs did not (Supplementary Fig. 3B). This is consistent with a recent study[31] and suggests that endosomal escape is not triggered by EV-induced endosomal membrane damage.

**Fig. 3 Quantification of the EV-content cytosolic release. a** Principle of EV-content cytosolic release quantification: signal from EV-content NLuc-CD63 (membrane marker) or NLuc-Hsp70 (cytosolic marker) should be associated with the membrane fraction (Mb), except if content release occurs. Then, only NLuHsp70 should be detected in the cytosolic fraction (C). **b** Scheme representing detergent-free cell fractionation protocol that separates membrane and cytosol fractions. Sonication and differential centrifugation allowed to fractionate the cells in three fractions: intact cell (Cells), membrane (Mb), and cytosolic (Cyt) fractions. **c** Immunoblots showing the distribution of Calnexin (ER luminal protein used to test for organelle integrity), CD63 (membrane marker), and Hsp70 (cystosolic marker) within each fraction emanating from HeLa WT cells that were intact (PBS), mechanically disrupted (Sonication), or detergent-disrupted (Triton-X-100). This blot is representative of four independent experiments. **d** Densitometry analysis of intact cell (black), membrane (dark gray), and cytosolic (light gray) fractions from sonication treatment in (**c**). From left to right $n = 6, 5, 5$. Error bars represent standard deviations. **e** NLuc activity quantification within intact cell (black), membrane (dark gray), and cytosolic (light gray) fractions. Stable HeLa NLuc-Hsp70 cells and transient HeLa NLuc-CD63 cells were submitted to detergent-free fractionation and luciferase activity was measured in each fraction, from left to right $n = 3, 4$. Error bars represent standard deviations. **f** Quantification of each marker in the cytosolic fraction from NLuc or western-blot (WB) quantification, after sonication (black) or Triton-X-100 (gray) treatment. Dashed lines represent the full range of experimentally measured maximum and minimal values. Each dot is an independent replicate and represents the mean of two technical replicates, $n$ from 2 to 5. Error bars represent standard deviations. **g** Quantification of the EV-content release: NLuc-CD63 or NLuc-Hsp70 EVs were loaded on unlabeled acceptor HeLa WT cells at 37 °C for 1–4 h, prior to performing the detergent-free cell fractionation protocol and measuring the Nluc activity to determine the % of each marker in the cytosolic fraction. As internal control, we measured the distribution of each marker within the donor cells. Dashed line represents experimental minimum and maximum values established in (**e**). Each dot is an independent experiment and represents the means of 2 technical replicates, $n = 4$. Error bars represent standard deviations. One-way ANOVA was performed with a Turkey's multiple comparison test. Indicated *p*-values, from left to right: 0.0091, 0.7963, and 0.0019.

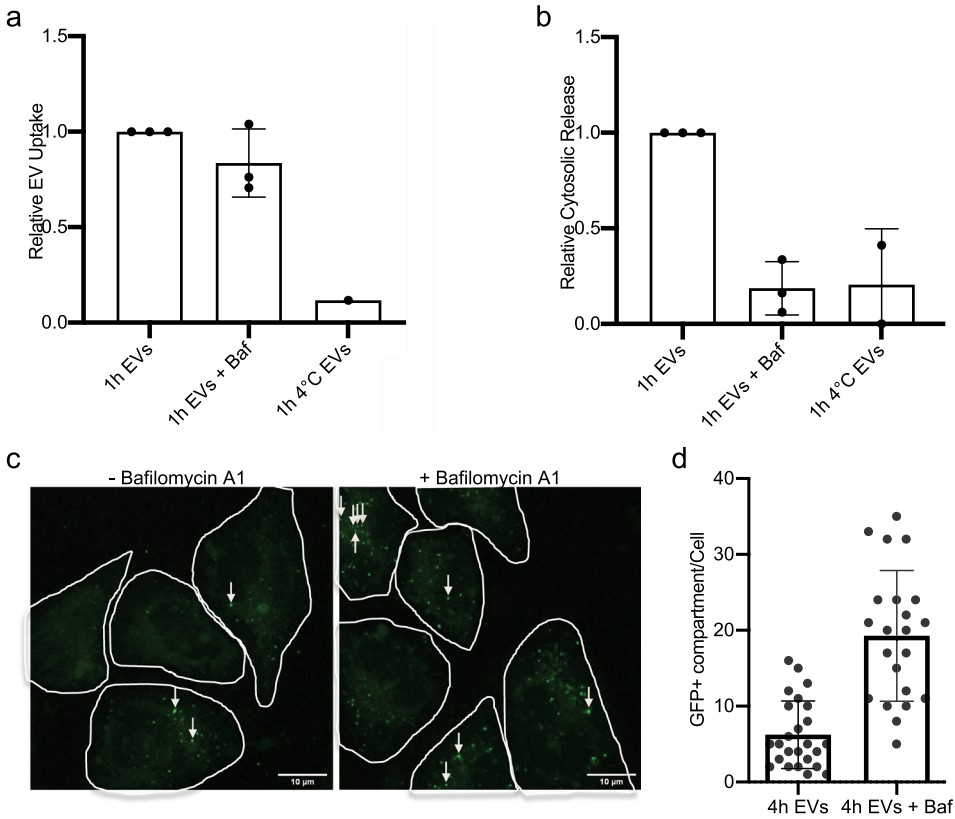

**Fig. 4 Endo/lysosomal acidification is required for EV-content delivery. a** Quantification of NLuc-Hsp70 EV uptake by unlabeled acceptor HeLa WT cells, with or without Bafilomycin A1 (Baf) treatment. Cells were incubated at 4 °C as negative control (no EV uptake). The EV uptake in control condition (non-treated) was set to 1. Each dot is an independent replicate and represents the means of 2 or 3 technical replicates, from left to right $n = 3, 3, 1$. Error bars represent standard deviations. **b** Quantification of NLuc-Hsp70 EV-content delivery within unlabeled acceptor HeLa WT cells, with or without Bafilomycin A1 (Baf) treatment. Cells were incubated at 4 °C as negative control (no EV-content release). The cytosolic release from the control condition (non-treated) was set to 1. Baf treatment (loss of endosomal acidification) inhibits the content release. Each dot is an independent replicate and represents the mean of 2 or 3 technical replicates, from left to right $n = 3, 3, 2$. Error bars represent standard deviations. **c** Confocal micrographs of HeLa acceptor cells after 4 h incubation with GFP-Hsp70 EVs with or without Baf treatment. White arrows indicate GFP-positive dots. Micrographs representative of three independent experiments. **d** Quantification of the number of GFP-positive dots per cell on unlabeled acceptor HeLa WT cells after 4 h incubation with GFP-Hsp70 EVs with or without Baf treatment. Each dot represents the value of GFP foci number within one acceptor cell, from left to right $n = 25, 22$. Error bars represent standard deviations.

**IFITM proteins inhibit EV-content release.** Aforementioned results and our cell-free study suggest that EV-content release may rely on pH-dependent fusion between EV and endosome membranes. Interestingly, a family of proteins called IFITMs has been shown to broadly inhibit fusion reactions during the processes of virus infection or syncytium formation[32–34]. We hypothesized that IFITMs may inhibit EV delivery if they used similar fusion mechanism.

We first engineered HEK293T cells that stably expressed flag-tagged IFITM1 and IFITM3. We used these cells because parental HEK293T are IFITMs negative[33,35]. We focused on IFITM1 known to be localized at the plasma membrane and to some extent at endo/lysosomal compartments, whereas IFITM3 is thought to show the mirrored distribution (endo > PM).

We incubated NLuc-Hsp70 positive EVs on control HEK293T cells or on cells overexpressing Flag-tagged IFITM1 or 3. Remarkably, EV-content delivery was inhibited (>60%) by both IFITM1 and 3, while the luciferase activity was slightly increased within the IFITM-positive acceptor cells (Fig. 5a and b). We confirmed through confocal microscopy that both IFITM proteins were localized at the plasma membrane and internal compartments (i.e., endo/lysosomes), with IFITM3 being more prominent than IFITM1 within those structures (Fig. 5c). We observed GFP-positive EVs within the vicinity of the plasma membrane, and internalized GFP-positive EVs (Fig. 5c). Most of those Internalized GFP-positive EVs were positive for IFITM1 (84%) or IFITM3 (92%) (Fig. 5c, d and Supplementary Fig. 4A). Closer analysis of the micrographs revealed that GFP foci either perfectly colocalized with IFITMs positive structures or were engulfed within IFITMs positive structures (Supplementary Fig. 4A). Only few GFP EV were negative for IFITMs (Supplementary Fig. 4A).

This suggests that IFITM1 or IFITM3 may sequester EVs within IFITMs positive endosomes, perhaps by blocking fusion. To test this hypothesis more directly, we used our cell-free assay that reconstitutes the content delivery step[14]. In this assay, EVs are incubated with plasma membrane sheets and proteinase K. At acidic pH, EV cargo (GFP-Hsp70) that is normally protected from proteinase digestion is now degraded (Fig. 5e–g samples 1–3). pH-dependent content release was again evidenced when PM sheets purified from parental HEK (IFITM negative) were mixed with EVs (Fig. 5f, g, samples 4, 5). Remarkably, presence of IFITM1 or 3 within the PM sheet abolished EV-content release (Fig. 5f and g, samples 6–9). PM sheet quality was tested and validated by western blot as described previously[14] (Supplementary Fig. 4B, C).

These results validate our hypothesis and suggest that IFITMs inhibit EV-content release by blocking membrane fusion between EV and target membranes.

## Discussion

We previously showed in vitro that EV-content release is pH-dependent[14]. Here we report that EV uptake and delivery within live cells is a low-yield process, which we quantified for the first time. We estimated that EV internalization occurs at a 1% spontaneous rate at 1 h, with up to 30% of those internalized EVs capable of content delivery. EV uptake is not saturated even at high doses of EV (>100 μg/ml). EV association with acceptor cells is inhibited at low temperature, consistently with previous reports[16]. In comparison, receptor-dependent LDL uptake is saturated at lower doses (<10 μg/ml) and LDL association with cell surface is not inhibited at 4 °C[36]. This suggests that EV uptake is not mediated by a bona fide receptor, at least within the tested cells. However, lack of specific receptors cannot be generalized yet, and it is possible that certain combinations of donor/acceptor

cells that communicate more efficiently through EVs use such receptors to increase EV targeting and capture.

Consistent with previous studies[12,18,37], we show that internalized EVs co-localize with endosomes and lysosomes to some extent. Importantly we showed that EV-content release is inhibited by bafilomycin A1 treatment, demonstrating again the pH-dependency of the process.

IFITMs are known to block the entry of several viruses within the cells[38]. It has been proposed that IFITM1, which is mainly at the plasma membrane, and IFITM3, which is mainly at late endosomes/lysosomes, differently impact viruses that use pH-independent or pH-dependent fusion to deliver their content into the cells[39]. Based on our results that support acidic endosomes as the point of delivery, and following this simple IFITM/location-based restriction rule, one could have predicted that IFITM3 should be more efficient to perturb EV-content release and might find our IFITM1-related results contradictory. However, in our hands, both overexpressed IFITMs are significantly localized at both PM and internal compartments (i.e., endosomes), although IFITM3 is more abundant within the latter. Importantly, we observed that most of the internalized EVs colocalized with or are sequestered within IFITM1 or IFITM3 positive compartments, where IFTIMs may restrict EV-content delivery. Interestingly, careful analysis of the literature on virus restriction entry inhibition by IFITMs revealed that IFITM specificity is not as exclusive as initially proposed[38,39]. In many cases, most IFITM proteins (including IFITM1) indeed restrict entry of virus that uses pH-dependent entry. Importantly, we independently confirmed with the cell-free assay the negative effect of both IFIMT1 and 3 on pH-dependent EV delivery.

Thus, the simplest interpretation of our data led us to propose that IFITMs (1 and 3) may block fusion between EVs and endosomal membranes. A role for IFITMs as negative regulator of membrane fusion is generally accepted, but the precise mode of action is still obscure[39]. IFITMs might perturb membrane fluidity, lipid distribution, and size extension of the fusion pore. Further investigation will be required to formally demonstrate that membrane fusion is a pre-requisite for EV-content delivery and clarify the mechanism by which IFITMs perturb such a fusion. For now, we propose that EVs are nonspecifically internalized within the cells, and that endosomal acidification triggers EV-content release, likely through a membrane fusion reaction.

How can such a low-yield process be physiologically relevant?

First, several viruses have similar low potency to enter into the cell, but benefit from the viral replication to amplify the infectivity[40].

Second, acute spatio-temporal coordination of a 1% yield process may still be compatible with several physiologically relevant functions. For instance, numerous spermatozoids go through the epididymis during their maturation, and it has been shown that RNA contained in EVs emanating from distal part of the epididymis can rescue unfertile sperms collected from upstream tissues[41,42]. It seems conceivable that a 1% efficient EV-mediated RNA transfer into sperms transiting through the epididymis could be sufficient to transmit selective advantage to the few beneficiary acceptor spermatozoids, to the detriment of unfertile sperm and allow efficient fecundation.

Third, functions relying on signal amplification may also be compatible with a 1% yield process. EVs have been shown to deliver antigens into dendritic cells capable to activate antigen-specific T-cell response[43]. Perhaps the delivery of an antigen within a dosage range compatible with the yield reported here, would be sufficient to efficiently presents this antigen at the surface of dendritic cells. These cells could then amplify the response through multiple contacts with numerous T cells[44], which once activated would trigger an efficient immune-response.

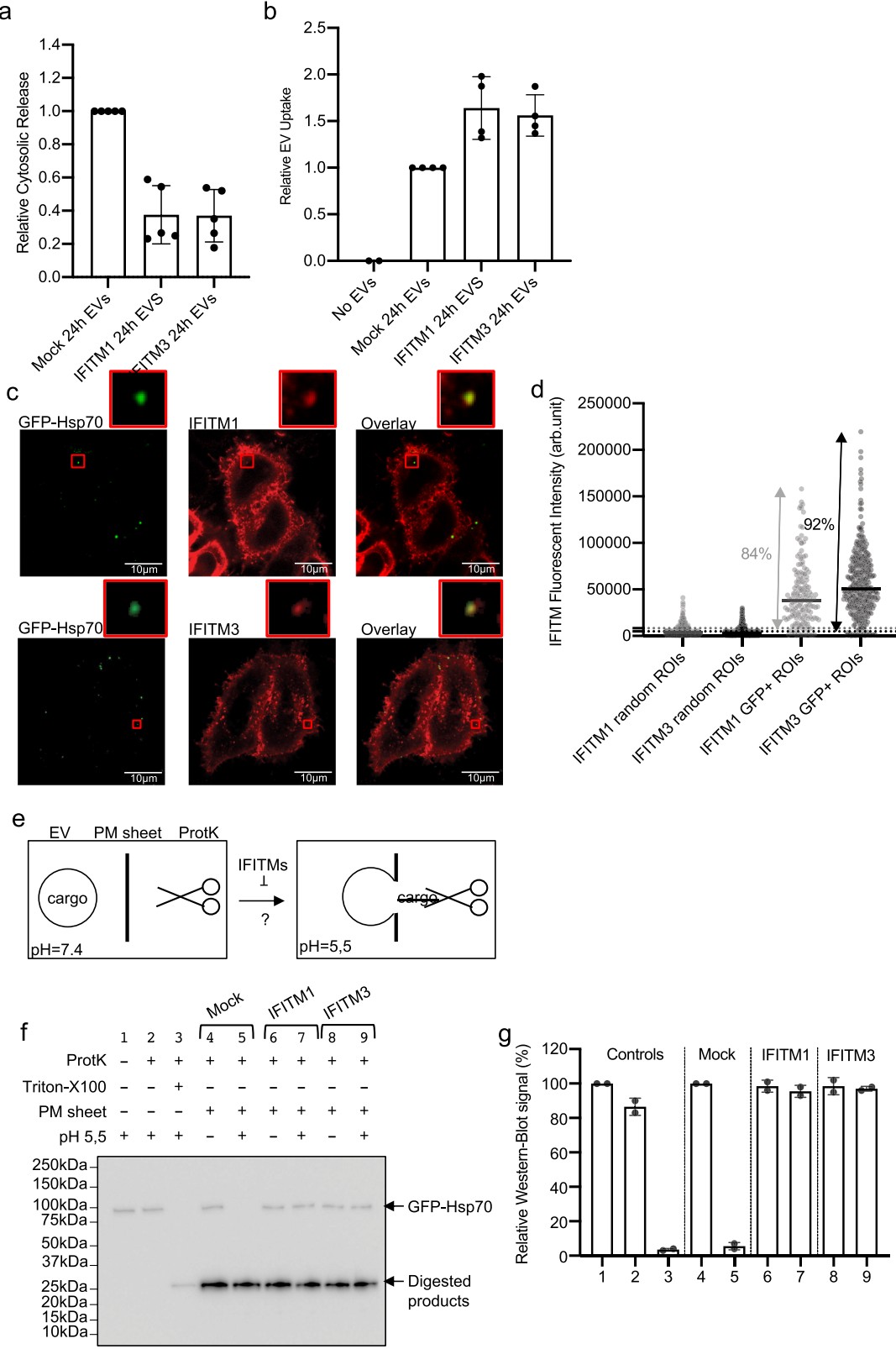

Other aspects of EV-mediated protein transfer that have been reported are more difficult to reconcile with <1% yield process. For instance, EV transport has been proposed to replenish in vivo adipocytes depleted of caveolin[45]. This would surely require a more efficient mechanism involving specific targeting and EV capture, that is not supported by our results. Perhaps certain tissues/cell combinations[45,46] are optimized for EV exchange and may help to identify a specific uptake machinery.

Finally, we showed in our system that the rate-limiting step is the uptake process (~1%). The content delivery process

**Fig. 5 IFITM proteins inhibit EV-content delivery. a** Quantification of NLuc-Hsp70 EV-content delivery within Mock HEK cells (control), or HEK cells stably expressing Flag-tagged IFITM1 or IFITM3. NLuc-Hsp70 EVs were loaded on each acceptor cell prior to performing cell fractionation and determining the portion of NLuc-Hsp70 found in the cytosolic fraction. Cytosolic release measured in control (Mock cells) was set to 1. Each dot is an independent replicate and represents the means of 2 or 3 technical replicates, $n = 5$. Error bars represent standard deviations. **b** Quantification of NLuc-Hsp70 EV uptake by control HEK cells (Mock), or stably expressing IFITM1, IFITM3. EV uptake measured in control (Mock) was set to 1. Each dot is an independent replicate and represents the means of 2 or 3 technical replicates, $n = 4$. Error bars represent standard deviations. **c** Confocal micrographs of IFITM1 or IFIMT3-HEK acceptor cells after uptake of GFP-Hsp70 EVs. IFITM1 and 3 were detected through their flag-tag (Alexa 546) whereas EV were detected through their GFP signal. Red squares show higher magnification of internalized EVs colocalizing with IFITM1 and 3. Micrographs representative of at least three independent experiments. **d** Quantification of Flag-IFITM1 (light gray) and 3 (dark gray) fluorescent signals in GFP+ ROI versus random ROI on HEK acceptor cells after incubation with GFP-Hsp70 EVs. One dot represents one ROI. $n = 241$ (control IFITM1), 387 (control IFITM3), 166 (GFP+IFITM1), 370 (GFP+IFITM3). Dashed lines represent the average background (random ROI) value plus twofold standard deviation. **e** Scheme illustrating the principle of the cell-free content EV-content release assay. Under acidic pH and in presence of PM sheets, EVs released their content (GFPHsp70) in the buffer where the cargo can be digested by proteinase K. The role of IFIMT1 and 2 in target PM sheets can be tested following this protocol. **f** Immunoblot showing the resistance of GFP-Hsp70 to proteinase K when EVs are incubated with target PM sheets that contain or not IFIMT1 and 3, and that are exposed or not to acidic pH. Digested products of GFP-Hsp70 and GFP-PM sheets (also observable on Supplementary Fig. 3C) show similar profiles. This blot is representative of two independent experiments. **g** Quantification of the GFP-Hsp70 signal (proteinase K resistant) by densitometry analysis. Each dot represents an independent replicate, $n = 2$, control (line 1) was set to 100%. Error bars represent standard deviations.

(~20–30%) is more efficient and suggests that donor and acceptor membranes used here can be further used to attempt to discover the putative pH-dependent fusion machinery.

## Methods

**Cell culture**. HeLa (from ATCC, Virginia, USA) and HEK293T (kindly received from O. Schwartz group, derived from parental HEK293T purchased at ATCC) cells were cultured in DMEM (Gibco, Illinois, USA) complemented with 10% FBS (Gibco, Illinois, USA or Biosera, France), at 37 °C 5% $CO_2$. HeLa GFP-Hsp70 or NLuc-Hsp70 stable cell lines were selected with geneticin 10 µg/mL (Gibco, Illinois, USA) after lipofectamine-based transfection. HeLa NLuc-CD63 was analyzed 1 day after transient transfection. HEK Mock, IFITM1 or IFITM3 stable cell lines were selected with puromycin 1 µg/mL (Gibco, Illinois, USA), after viral transduction. Except mentioned otherwise, cells were transfected using Lipofectamine 2000 (Invitrogen, Massachusetts, USA), according to manufacturer's protocol.

**Plasmids**. NLuc-CD63 construct was obtained by removing RFP sequence RFP-CD63 (a gift from Walther Mothes), using AgeI and XhoI restriction enzyme (NEB, Massachusetts, USA). PCR amplified (Supplementary Table 1) NLuc sequence (Promega, Wisconsin, USA) was then inserted using the same enzymes. NLuc-Hsp70 was obtained by replacing GFP sequence from GFP-Hsp70 (15215, Addgene, Massachusetts, USA) by NLuc sequence from NLuc-CD63 plasmid. Here again, AgeI and XhoI were used to digest both plasmids before ligation. CD8-ENLYFQS-GFP, used to generate PM sheets[14].

**EV isolation**. Donor cells were cultured for 24 h in serum-free DMEM. Conditioned media was harvested and submitted to a 2000 × g centrifugation for 20 min at 4 °C to remove cell debris, and then to a 100,000 × g ultracentrifugation for 1 h 30 min at 4 °C (45 Ti rotor and Optima L-80 ultracentrifuge, Beckman Coulter, California, USA). Note that in initial experiments, intermediate step at 10,000 × g was performed to remove large vesicles and large protein aggregates, but this fraction was virtually depleted of proteins including EV cargo of interest (GFP-Hsp70 and NLuc-Hsp70). We subsequently remove this additional step. The obtained pellet was washed with PBS and centrifuged 1 h at 100,000 × g 4 °C (MLA 80 rotor and Optima MAX-XP ultracentrifuge, Beckman Coulter, California, USA). Pellet was resuspended in 100 µL PBS and either stored at 4 °C (for up to 20 h) or immediately applied on acceptor cells.

For floatation assays, we proceeded as follow[28]. Briefly, 100,000 × g pellet obtained as aforementioned pellets were resuspended in 1 ml 60% sucrose and deposited in the bottom of the tube. One milliliter of 30% sucrose solution and 1 ml of PBS were sequentially loaded on top of the samples. Samples were centrifuged at 150,000 × g for 16 h at 4 °C (SW55 rotor). One milliliter of each fraction was collected and mixed with 6 ml PBS to dilute sucrose and eliminate its putative interference within our assays. Samples were centrifuged at 150,000 × g for 1 h 30 min (MLA 80 rotor), supernatant containing sucrose was removed and pellets were resuspended in 100 µL PBS prior to further testing for NLuc activity, protease protection assay, or uptake assay.

**Western blot and In-Gel NLuc detection**. Cells were resuspended in lysis buffer (Tris 50 mM, NaCl 150 mM, Triton-X-100 1%, protease/phosphatase inhibitor (PPI) cocktail (Roche, Switzerland), pH 8) 30 min on ice, then samples were submitted to 20 min 19,000 × g centrifugation. Supernatants (cell lysates) were collected. Protein concentration of cell lysate (CL) and EV samples were obtained using Micro BCA™ Protein Assay kit (Thermo Scientific, Illinois, USA). Samples were mixed with 4X Laemmli buffer (Bio-Rad, California, USA) 10% β-mercaptoethanol, except for In-Gel, CD63, and CD9 detection (no β-mercaptoethanol) and loaded on 4–15% polyacrylamide gels (Bio-Rad, California, USA). After electrophoresis, proteins were transferred on PVDF membranes using the Trans-Blot Turbo system (Bio-Rad, California, USA). Membranes were blocked in PBS 0.05% Tween20 5% milk, then incubated with 1/1000 primary antibody (α-Actin (MAB1501, Millipore, Germany), α-ALIX (clone3A9, 2171, Cell Signaling, Massachusetts, USA), α-Calnexin (ab133615, Abcam, UK), α-CD63 (clone H5C6, 556019, BD Bioscience, New Jersey, USA), α-CD9 (clone MM2/57, cbl162, Millipore, Germany), α-Hsp70 (ADI-SPA-810-D, Enzo LifeScience, New York, USA), α-FLAG (F3165, Sigma-Aldrich, Missouri, USA), α-GM130 (ab32337, Abcam, UK)) in PBS 0.05% Tween20 5% milk, washed and finally incubated with 1/5000 HRP-coupled secondary antibody (α-mouse or α-rabbit, 115-035-003, Jackson ImmunoResearch, UK) in PBS 0.05% Tween20. Membranes were developed using BM Chemiluminescence Western blotting Substrate (POD) (Roche, Switzerland) and ChemiDoc imager (Bio-Rad, California, USA). For the In-Gel detection of the NLuc, the same procedure was used until the migration of the proteins in the polyacrylamide gel. Then, the gel was handled as recommended by the supplier of the kit Nano-Glo™ In-Gel Detection System (Promega, Wisconsin, USA). For both methods, image analysis and quantification were performed using Image Lab Software (Bio-Rad, California, USA) and ImageJ software.

**Proteinase K protection assay**. Isolated NLuc-Hsp70 or NLuc-CD63 EV samples were incubated in PBS with or without 0.1% Triton X-100 and 50 µg/mL Proteinase K (AM2542, Ambion, Texas, USA) for 5 h at RT within 96-well plate. Then 20 µL of Nano-Glo™ Live Cell assay reagent (N2011, Promega, Wisconsin, USA) was added on each sample to immediately measure the remaining NLuc activity in each well using iD3 SpectraMax microplate reader (Molecular Devices, California, USA). For Alix resistance assessment by immunoblot, proteinase K was neutralized by adding 0.2 µL of boiled 0.2 µM PMSF immediately followed by loading buffer (100 °C, 15 min)[14].

**NLuc EV uptake**. Acceptor cells were seeded 24 h before the uptake experiment, at 20,000 cells per well in a 96-well plate. NLuc-Hsp70 EV input was added in serum-free DMEM for a final concentration of 1–10 µg protein/mL, or as specifically indicated. Cells were incubated with EVs at 4 or 37 °C for 0–24 h, or the specifically indicated time. After incubation, cells were washed with PBS and then lysed in the aforementioned lysis buffer, prior to transferring in white 96-well plates. Finally, 50 µL of Nano-Glo™ reagent (Promega, Wisconsin, USA) was added on each well and luminescence activity was read using iD3 SpectraMax microplate reader (Molecular Devices, California, USA) or Centro LB 960 microplate luminometer (Berthold, Germany) (for initial experiments related to Figs. 1b and 2a).

**GFP-EV uptake**. Acceptor cells were seeded 24 h before the uptake experiment, at 200,000 cells per well in a 24-well plate on the top of coverslips. For HEK cells, we used coverslips coated with poly-L-lysine (P8920, Sigma-Aldrich, Missouri, USA). Acceptor cells were incubated with GFP-Hsp70 EVs (10 µg protein/mL) for 1 h to 24 h at 37 °C. For Galectin3 labeling experiments, cells were treated 30 min with 500 µM LLOME (L7393-500MG, Sigma-Aldrich, Missouri, USA). Then cells were washed with PBS, fixed 15 min RT with PBS 4% Paraformaldehyde and permeabilized 30 min RT in PBS 1% bovine serum albumin (BSA) 0.1% Triton-X-100. Primary antibodies (α-Rab5 (610724, BD Bioscience, New Jersey, USA), α-Lamp1 (GTX62434, GeneTex, California, USA) or α-Galectine3 (14-5301-82, eBioscience, California, USA), or α-FLAG (F3165, Sigma-Aldrich, Missouri, USA)) were incubated ON at 4 °C at 1/100 in PBS 1% BSA 0.1% Triton-X-100. After washes in

PBS, samples were incubated 1 h RT with secondary antibodies (α-mouse or α-rabbit coupled to AlexaFluor (AF) 488, 546, or 647 nm (ThermoFisher, Massachusetts, USA)) at 1/500 in PBS. Once washed, samples were mounted and image using a SP8 confocal microscope (Leica Microsystems, Germany). Image analysis and colocalization quantification were performed using ImageJ software (NIH, Maryland, USA). Briefly, to assess compartment colocalization, the fluorescent signal coming from acceptor cell protein in region of interest (ROI) corresponding to EV surface for Rab5 and Lamp1, or to 5 μm² ROI centered on the EV signal.

**Acceptor cell fractionation.** Acceptor cells were seeded 24 h before the uptake experiment, at 200,000 cells per well in a 24-well plate. The same procedure for NLuc EV uptake above was used for the incubation with NLuc-Hsp70 or NLuc-CD63 EVs. To assess the dependency on endosomal acidification, acceptor cells were treated or not with 200 nM of Bafilomycin A1 (SML1661, Sigma-Aldrich, Missouri, USA) from 30 min before adding EV until the end of the uptake assay. Then acceptor cells were washed with PBS, detached, and collected using PBS 0.5 mM EDTA, and pelleted 10 min at 350 × g 4 °C and resuspended in PBS 1X PPI. When required Triton-X-100 (positive controls) was added at that stage at 1% final concentration and samples were kept at 4 °C. For detergent-free cell disruption, samples were processed as follows: 5 s vortex, 5 back-and-forth in 30G needle and 5 s sonication (30% duty cycle, output control level 3) with a micro-tip sonicator (Ultrasonic Processor, Thomas Scientific, Swedesboro, NJ, USA).

Sample was then submitted for 10 min, 350 × g, 4 °C, to pellet the intact cells (resuspended in PBS 1X PPI), then the supernatant was centrifuged 1 h at 100,000 × g 4 °C to pellet the membranes (resuspended in PBS 1X PPI) and to recover the cytosolic fraction (supernatant). Distribution of the proteins in the different fractions was determined either by western-blot or NLuc activity measurement.

**Cell-free EV-content release assay.** HEK-derived PM sheets were prepared as follows. Briefly, control HEK cells (Mock) or cell stably expressing Flag-IFITM1 or 3 were transfected with plasmid encoding CD8-ENLYFQS-GFP. Two days after transfection, cells were washed in PBS, detached, and collected. After centrifugation (350 × g, 4 °C, 5 min), cells were resuspended in 400 μL PBS, prior to being incubated with 100 μL of Protein G-conjugated magnetic beads (Bio-Rad, Hercules, CA, USA) and 2 μL of anti-CD8 antibody (clone Rpa-T8; eBioscience, San Diego, CA, USA) at 4 °C for at least 4 h. Non-attached cells were removed and bead-attached cells were submitted to sonication in 400 μL ice-cold PBS using a micro-tip sonicator (Ultrasonic Processor, Thomas Scientific, Swedesboro, NJ, USA; three pulses of 5 s, 30% duty cycle, output control level 3). Supernatant (cytoplasm) was discarded, PM remaining attached to the beads were washed with PBS. Bead-attached membranes were resuspended in 100 μL PBS using TEVp (Sigma-Aldrich, Missouri, USA) at 1 μg/mL, overnight at 4 °C on rotative wheel. Supernatant containing released-PM sheet was collected and protein concentration was determined using Micro BCA protein assay kit prior to testing PM sheet quality by western blot.

GFP-Hsp70 containing EVs were preincubated or not with HEK-derived PM sheets (1:1 protein ratio) at 4 °C for 1 h prior incubation at 25 °C for 1 h. When required, samples were treated with 1% Triton X-100. When required we added the 'fusion' buffer (10 mM Na$_2$HPO$_4$, 10 mM NaH$_2$PO$_4$, 150 mM NaCl, 10 mM 2-(N morpholino) ethanesulfonic acid, 10 mM N-2-hydroxyethylpiperazine-N 9-2-ethanesulfonic acid, adjusted to pH 5.5 or required pH) for 1 h at 25 °C. Then samples were incubated at 4 °C with proteinase K (2 μg/mL) for 1 h. Proteinase K was inactivated as for protease protection assay above.

**Statistical analysis.** First, normality of data was assessed using Shapiro–Wilk test, and Levene's test was used for the homogeneity of variance. When positive, one-way ANOVA was performed (Fig. 3g).

**Reporting summary.** Further information on research design is available in the Nature Research Reporting Summary linked to this article.

## Data availability

The authors declare that the data supporting the findings of this study are available within the Supplementary information files. Further requests should be addressed to the corresponding author. Source data are provided with this paper.

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

## Acknowledgements

We thank Mathieu Maurin for his input on image quantification. We thank Nicolas Manel for his suggestion concerning the IFITMs. We thank Alena Ivanova and Michael Boutros for introducing us to NanoLuc Luciferase. We thank Marine Gros for her help with Galectin3-related experiments. E.B. is a recipient of a PhD fellowship from La ligue contre le Cancer. E.G. is a recipient of a post-doctoral fellowship from Fondation pour la Recherche Médicale (FRM (FRM: SPF20170938694). This work is supported by Fondation ARC (PJA20171206453 and PGA1 RF20180206962), Cancéropôle Île-de-France (Emergence 2018) and French National Research Agency (ANR-10-IDEX-0001-02 PSL*, ANR-11-LABX-0043, and ANR-18-CE15-0008, ANR-19-CE18-002003), and ANR-18-IDEX-0001, IdEx Université de Paris. OS lab is funded by Institut Pasteur, ANRS, Sidaction, the Vaccine Research Institute (ANR-10-LABX-77), Labex IBEID (ANR-10-LABX-62-IBEID), "TIMTAMDEN" ANR-14-CE14-0029, "CHIKV-Viro-Immuno" ANR-14-CE14-0015-01, ANR/FRM Covid support, and the Gilead HIV cure program.

## Author contributions

E.B. and G.L. designed, performed, and analyzed the experiments. E.G. and J.B. generated molecular and cellular tools. E.B., E.G., J.B., O.S., C.T., and G.L. wrote or revised the manuscript.

## Competing interests

The authors declare no competing interests.
