## [Peer Review File · Nature Communications]

REVIEWER COMMENTS

Reviewer #1 (Remarks to the Author):

In this manuscript, Bonsergent et al apply biochemical methods to better understand the uptake of extracellular vesicles (EVs) and the delivery of the cargo into living cells. The latter is of particular importance at this time, as the EV field is grappling with the question of how cargo is utilised. We know more about how EVs get into cells, but to understand how the cargo is used (which is arguably the more important question) we require new methods such as the work presented here. Overall, I feel the work is of high quality and should be accepted with some corrections. I do have some suggestions/comments that should be addressed:

Fig 1c shows the characterisation of the EVs by western blotting. Ideally there would be an additional marker for an endogenous luminal component of EVs, such as Alix. The authors have already shown that the NLuc-Hsp70 is present, and although it is a modified and over-expressed protein I feel it's probably sufficient for this purpose. As key contributors of the MISEV 2018 guidelines, I know the authors will be well aware of the requirements for publishing on EVs; ideally it would be good to see characterisation by a second method other than western blotting, such as electron microscopy imaging, but to be honest I've no doubt these labs are showing us bona fide EVs and would leave it to the authors to decide if this additional characterisation is necessary.

The experiments to measure whether the NLuc activity is within EVs (fig1D) is important. The results reveal that the most of the protein appears to be vesicular. Nevertheless, approximately 20-30% of the signal appears to be lost after PK treatment without Triton, suggesting some of the protein may be non-vesicular. Given that 30% of the uptaken signal is released into the cytoplasm (fig3G), is it possible that this fraction coincides with the non-vesicular Hsp70? Perhaps the authors could comment? As a control the authors should show that proteinase K treatment of the EV sample does not affect the 30% release of hsp70 luc into the cytoplasm shown in Fig3G? Alternatively (and if the proteinase K affects the ability of EVs to fuse with internal membranes, which may be an interesting experiment in of itself), fig1D and 3G could be repeated on EVs that have been purified much more extensively (e.g. sucrose gradient).

In the EV-uptake characterisation section, the authors state that "The luciferase activity that is normally associated with acceptor cells after 1h of EV treatment was virtually absent in cells treated at 4°C, suggesting the lack of EV binding to the acceptor cell surface via a bona fide receptor (Figure 2 C)." I'm not convinced there are enough data here to reach this conclusion about receptors. The authors are quantifying total cellular accumulation of hsp70-NLuc, and as far as I can tell absence of signal in the assay does not preclude the possibility of surface-associated EVs that are interacting with receptors but unable to internalised due to the non-permissive temperature. I feel the authors should either tone this statement down, or perform additional control experiments to provide stronger evidence for the claim.

The data in Figure 3G are exciting. They suggest that in recipient cells the NLuc-Hsp70 is released from EVs into the cytoplasm. I would like to see one further control. Can the authors be certain that treatment of the recipient cells with EVs causes a destabilisation of the endosomal trafficking

compartment that results in loss of integrity and general transfer of luminal material (including uptaken EVs and their cargo) into the cytoplasm? To show this is specific transfer of cargo I think the same cells in fig3G should be assessed for endogenous proteins to show that the normal distribution of luminal endosome proteins remain associated with the membrane fractions. I think it's likely to show that the cytoplasmic EV content is a specific processes and not a general loss of endosomal cohesion, but I feel it's important to formally rule it out.

Fig4D suggests a greater number of 'GFP compartments/cell' which is "consistent with EV confinement within neutralized endo/lysosomes". I don't understand why this is being used as a proxy for retention in the endosomal system. If Bafilomycin is preventing EVs from escaping the endosomal compartment then wouldn't the best way to quantify this be to demonstrate an increased co-localisation with late-endosomal or lysosomal markers..?

I particularly enjoyed the discussion of the compatibility of a 1%/30% efficient uptake/release process with physiological functions.

Reviewer #2 (Remarks to the Author):

In this manuscript, Bonsergent et al present a cell-based assay that they developed to quantitatively determine the efficacy of extracellular vesicle (EV) uptake and content release in acceptor cells. The assay employs tagging of EV-cargo with luciferase or GFP, which is then quantified and visualized in acceptor cells under different experimental conditions. The study addresses a highly relevant topic in the EV-field, as EV-cargo release and its access to the cytosol of acceptor cells is mechanistically not understood and difficult to quantify. The manuscript is concise and provides new insight in EV-uptake and release kinetics and furthermore suggests that EV fusion occurs within acidified endosomes. In my opinion, the study is well designed, innovative and of potential value to the broader field. However, it suffers from certain limitations detailed below, the authors may be able to address.

Major points:

1. The study focusses on Hsp70 as cytosolic cargo and homotypic fusion of HELA-derived EVs with HELA acceptor cells. This limits the interpretation of the results largely to tumor cell communication and does not necessarily apply to heterotypic EV-mediated communication in a physiological (and pathological) context. The conclusions regarding receptor-independent uptake, efficacy of cargo uptake and release may be very different in these contexts. In my opinion, this should be better reflected and discussed more extensively.
2. Besides technical validation of the assay, the study presents kinetics pointing out numbers such as 1 % uptake and 30 % content release rate (which are generalized to exhibit broader validity). However, the number of biological replicates underlying these data is not sufficient to put these numbers on a solid statistical basis (e.g. n=2 for uptake assays; n=4 for content release assay, which collects two different time points of analysis). The density of data needs to be increased to state these numbers as a general finding of the study.

3. Although there is no doubt that Hsp70 is EV-associated, there are several indications in the literature that it can leave the cell via other routes. Fig. 1 D indeed shows that protease digestion of EVs reduces the Hsp70 cargo by about 25 %, suggesting that a significant part of Hsp70 in the 100.000 x g pellet is not membrane-protected and appears to be present outside EVs. The level of EV-association of Hsp70 could be further explored using density gradient centrifugation or size exclusion chromatography. Analysis of other typical cytosolic EV-cargoes, such as Alix or syntenin, would further increase the body of evidence and verify the results (or reveal that there are other EV-subtypes with distinct uptake and cargo-release kinetics, which would not be surprising). However, I see that analysis of other cargoes in the assay may be beyond the scope of the present manuscript.

4. The study compares CD63 as a membrane-associated and Hsp70 as internal EV cargo. Cargo release is measured only for Hsp70 in the cytosolic fraction and ignored for CD63, which actually could be measured in the membrane fraction. In the turn of membrane fusion, CD63 is incorporated in the cellular membrane pool and will become protease sensitive (while non-fused EVs would be hidden in the endosomal compartment and protease-resistant). The relation between total membrane-associated NLuc-CD63 and protease-sensitive NLuc-CD63 would give information on the release rate of CD63. Have the authors performed such experiments? Furthermore, Fig. 3G indicates that 10 % of NLuc-CD63 can be recovered from the cytosol. Is this due to membrane background in the cytosolic fraction and does this finally mean that the 30 % recovery rate should be corrected by this background value?

5. Based on their results, can the authors exclude fusion at the plasma membrane? The requirement of acidification for content release may be a strong argument for endosomal release, but how about local pH-changes at the plasma membrane? Could these pH changes be sufficient to drive EV-fusion?

Minor points:

1. Fig. 3 G, quantification of EV-content release: The data are expressed relative to the total activity measured in the fractions. It would be informative to see how this relates to the total activity loaded to the cell. Furthermore, the data are collected from different timepoints of EV uptake (1-4 h). It would actually be interesting to see kinetics and how cytosolic NLuc-Hsp70 develops with time.

2. Fig. 2E: it is not clear how the measurement was performed (please explain in more detail at least in the methods) and what the individual data points reflect (compartments within cells?). How many cells were analyzed in total? The text claims 30 % co-localization with early endosomes at 1h; where is this information reflected in the figure?

3. Methods/statistical analysis: please reconsider the application of student's t-test in the designated figures (number of n appears too low to claim normality of the data).

4. Methods: please explain acronym PPI.

Eva-Maria Krämer-Albers

We would like to thank the reviewers for the constructive comments. We hope that we have satisfyingly addressed all the key points, and that the manuscript now reaches the standards for publication.

Please find below our detailed response (blue ink).

Reviewer #1 (Remarks to the Author):

In this manuscript, Bonsergent et al apply biochemical methods to better understand the uptake of extracellular vesicles (EVs) and the delivery of the cargo into living cells. The latter is of particular importance at this time, as the EV field is grappling with the question of how cargo is utilised. We know more about how EVs get into cells, but to understand how the cargo is used (which is arguably the more important question) we require new methods such as the work presented here. Overall, I feel the work is of high quality and should be accepted with some corrections. I do have some suggestions/comments that should be addressed:

Thanks for acknowledging the high quality and the importance of our work.

Fig 1c shows the characterisation of the EVs by western blotting. Ideally there would be an additional marker for an endogenous luminal component of EVs, such as Alix. The authors have already shown that the NLuc-Hsp70 is present, and although it is a modified and over-expressed protein I feel it's probably sufficient for this purpose.

We have now added Alix as an endogenous cytosolic marker in our study (figure 1C and Supplementary figure 1C and E). Its behavior (presence in EV fraction, resistance to protease degradation) is similar to the one we reported for the overexpressed cargo-Hsp70.

As key contributors of the MISEV 2018 guidelines, I know the authors will be well aware of the requirements for publishing on EVs; ideally it would be good to see characterisation by a second method other than western blotting, such as electron microscopy imaging, but to be honest I've no doubt these labs are showing us bona fide EVs and would leave it to the authors to decide if this additional characterisation is necessary.

The method used here to isolate EVs is similar to the one that we used in our recently published study (Bonsergent et al, 2019), in which we showed EM pictures and EV size quantifications. Below (figure 1) is another example of EM picture that I hope will completely erase the last suspicion of doubt from this reviewer. Note, that since those pictures are only

qualitative, we do not think that they bring strength to the manuscript and have decided to not include them in the manuscript.

Figure 1 Electron micrograph showing Isolated EVs (100 000g pellet)

The experiments to measure whether the NLuc activity is within EVs (fig1D) is important. The results reveal that the most of the protein appears to be vesicular. Nevertheless, approximately 20-30% of the signal appears to be lost after PK treatment without Triton, suggesting some of the protein may be non-vesicular.

It is true that 25 +/-5 % of NLucHsp70 is sensitive to proteinase K. Importantly, we showed that it was very similar for Nluc-CD63 (20+/-11%), although it is a membrane-bound protein harboring the tag on the cytosolic face of the EV membrane. Importantly, and as suggested by both reviewers, we used floatation assay as a second method to isolate EVs. We observed the very same proteinase K sensitivities for NlucHsp70 and NlucCD63 (24 +/-4 % and 24 +/-7 %, respectively, supplementary figure 1D). We report similar results for endogenous Alix (supplementary Figure 1E). Our simplest explanation is that this modest but yet measurable unexpected degradation reflects EV membrane damage induced by the isolation procedure. This confirms our previous results and indicates that we are indeed dealing with cargo contained within vesicles

Given that 30% of the uptaken signal is released into the cytoplasm (fig3G), is it possible that this fraction coincides with the non-vesicular Hsp70? Perhaps the authors could comment? To address that comment, we loaded NLuc-Hsp70 containing EVs that were isolated through the floatation assay on acceptor cells. The uptake (1,2%+/-0,2) was similar to the one reported for EV isolated through ultracentrifugation alone (supplementary figure 1 F). This suggests again that we are indeed characterizing the fate of Hsp70 containing EVs and not free soluble NLucHsp70 that cannot float.

Importantly, we added an additional control. We loaded, on acceptor cells, recombinant free soluble NLuc and observed that almost none of the recombinant protein could be internalized by the cells (0,03% +/- 0,01).

Altogether the floatation method confirms all our results and strongly indicates that we are more than likely to follow the fate of cargo contained within EVs.

As a control the authors should show that proteinase K treatment of the EV sample does not affect the 30% release of hsp70 luc into the cytoplasm shown in Fig3G?

Unfortunately, this is not possible as Proteinase K treatments prevents EV content delivery, as demonstrated by our previous cell-free study (Bonsergent et al, 2019), which suggests that the delivery process requires proteins at the surface of both donor EV and target membranes. In addition, proteinase K treatment of acceptor cells trigger cell detachment, which prevented us to perform the uptake and content delivery assay with our present set-up.

Alternatively (and if the proteinase K affects the ability of EVs to fuse with internal membranes, which may be an interesting experiment in of itself), fig1D and 3G could be repeated on EVs that have been purified much more extensively (e.g. sucrose gradient). We followed this recommendation and showed that EV purified through floatation assay behave as EV isolated through ultracentrifugation alone, as mentioned above. Again, we believe that this second method of isolation confirms and strengthens our previous results.

We have included all these new results in a supplementary figure 1 and added a paragraph reporting the convergence of our data regardless of the method of EV isolation;

In the EV-uptake characterisation section, the authors state that “The luciferase activity that is normally associated with acceptor cells after 1h of EV treatment was virtually absent in cells treated at 4°C, suggesting the lack of EV binding to the acceptor cell surface via a bona fide receptor (Figure 2 C).” I’m not convinced there are enough data here to reach this conclusion about receptors. The authors are quantifying total cellular accumulation of hsp70-NLuc, and as far as I can tell absence of signal in the assay does not preclude the possibility of surface-associated EVs that are interacting with receptors but unable to internalised due to the non-permissive temperature. I feel the authors should either tone this statement down, or perform additional control experiments to provide stronger evidence for the claim.

Incubation at 4° C is indeed designed to assess if EVs can bind to the cell surface of acceptor cells, through protein-protein interaction that still occurs at that lower temperature. For instance, this classical experiment has been used in the past to characterize and identify ldl receptor that binds ldl (a “classic” of cell biology from Goldstein and Brown already cited in our manuscript). With our assay, if NLuc-positive EVs were to bind the cell surface of the acceptor cells, we should be able to quantify the luciferase activity associated with those cells at 4°C. This is not the case. As per request, we performed additional experiments and repeats (two more independent experiments, each including 2 technical replicates, and kinetics extended at 2 hours). We confirmed our previous results. Even after two hours of incubation at 4°C, the luciferase activity is lower than 0,4 % +/- 0,2, and virtually indistinguishable from the background signal. If a *bonafide* receptor was responsible for EV capture through protein-protein interaction, one would expect to detect significant NLuc activity, comparable to the one monitored at 37°C. This is not the case. In addition, we failed to image fluorescent EVs accumulating at the cell surface when cells were incubated at 4°C (data not shown). This led us to propose in our first version that EV uptake is not mediated by a *bonafide* receptor, at least in the cell lines that we tested. Reviewer 2 also expressed concerns about the generalization of this proposition. We agree that the apparent lack of *bonafide* receptor cannot be generalized yet. For that reason, we cautiously removed from the abstract the sentence containing “lack of bonafide receptor”. To further temper our statement, we now add a brief comment in the discussion section mentioning that “However, lack of specific receptors cannot be generalized yet, and it is possible that certain combinations of donor/acceptor cells that communicate more efficiently through EVs use such receptors to increase EV targeting and capture”.

The data in Figure 3G are exciting. They suggest that in recipient cells the NLuc-Hsp70 is released from EVs into the cytoplasm. I would like to see one further control. Can the authors be certain that treatment of the recipient cells with EVs causes a destabilisation of the endosomal trafficking compartment that results in loss of integrity and general transfer of luminal material (including uptaken EVs and their cargo) into the cytoplasm? To show this is specific transfer of cargo I think the same cells in fig3G should be assessed for endogenous proteins to show that the normal distribution of luminal endosome proteins remain associated with the membrane fractions. I think it’s likely to show that the cytoplasmic EV content is a specific processes and not a general loss of endosomal cohesion, but I feel it’s important to formally rule it out.

A recent paper, no cited in our manuscript, showed that EV uptake do not trigger endosomal damage (Joshi et al,2020, Acs nano). However, we tested this possibility and used antibody against galectin3 that recognized the luminal side of disrupted endosomes that is normally not accessible. As a control we used LLOME, a chemical agent that damages endosomes. Consistent with this paper we showed that LLOME disrupt endosomal integrity and colocalized with galectin3 signal, whereas uptaken EVs do not. This demonstrates that internalized-EVs do not disrupt endosomal membrane integrity. Together with the IFIMT-

related data, these results, now reported in supplementary figure 2A, suggest that content delivery is more likely to require membrane fusion than membrane disruption.

Fig4D suggests a greater number of 'GFP compartments/cell' which is "consistent with EV confinement within neutralized endo/lysosomes". I don't understand why this is being used as a proxy for retention in the endosomal system. If Bafilomycin is preventing EVs from escaping the endosomal compartment then wouldn't the best way to quantify this be to demonstrate an increased co-localisation with late-endosomal or lysosomal markers..? We thought that it was important to show that bafilomycin increases the number of GFP-foci to show that release of EV-cargo (GFP- Hsp70) is pH dependent. Therefore, the first step was to demonstrate that bafilomycin treatment triggered accumulation of internalized EVs. We agree with reviewer 1 that it is equally important to demonstrate that those GFP-foci also colocalize with endosomes unable to mature. We now show that all these numerous GFP foci do co-localize with Rab5 in cells treated with bafilomycin. This is consistent with the other data that strongly suggest that acidification is required for EV content release.

I particularly enjoyed the discussion of the compatibility of a 1%/30% efficient uptake/release process with physiological functions.

Thanks, I enjoyed writing it.

Reviewer #2 (Remarks to the Author):

In this manuscript, Bonsergent et al present a cell-based assay that they developed to quantitatively determine the efficacy of extracellular vesicle (EV) uptake and content release in acceptor cells. The assay employs tagging of EV-cargo with luciferase or GFP, which is then quantified and visualized in acceptor cells under different experimental conditions. The study addresses a highly relevant topic in the EV-field, as EV-cargo release and its access to the cytosol of acceptor cells is mechanistically not understood and difficult to quantify. The manuscript is concise and provides new insight in EV-uptake and release kinetics and furthermore suggests that EV fusion occurs within acidified endosomes. In my opinion, the study is well designed, innovative and of potential value to the broader field. However, it suffers from certain limitations detailed below, the authors may be able to address.

Thanks for the general good appreciation of our work.

Major points:

1. The study focusses on Hsp70 as cytosolic cargo and homotypic fusion of HELA-derived EVs with HELA acceptor cells. This limits the interpretation of the results largely to tumor cell communication and does not necessarily apply to heterotypic EV-mediated communication in a physiological (and pathological) context. The conclusions regarding receptor-independent uptake, efficacy of cargo uptake and release may be very different in these contexts. In my opinion, this should be better reflected and discussed more extensively.

Actually, we also used HEK cells in the second part of the study, when dealing with IFITM proteins, and results were comparable. However, we agree that the apparent lack of *bonafide* receptor cannot be generalized. As mentioned in response to reviewer 1, we cautiously removed, from the abstract, the sentence containing "lack of *bonafide* receptor". Note that in the original manuscript we already cautiously proposed that "EV uptake is not mediated by a bona fide receptor, at least within the tested cells". To further temper our statement, we now add a brief comment "However, lack of specific receptors cannot be generalized yet, and it is possible that certain combinations of donor/acceptor cells that communicate more efficiently through EVs use such receptors to increase EV targeting and capture".

2. Besides technical validation of the assay, the study presents kinetics pointing out numbers such as 1 % uptake and 30 % content release rate (which are generalized to exhibit broader validity). However, the number of biological replicates underlying these data is not sufficient to put these numbers on a solid statistical basis (e.g. n=2 for uptake assays; n=4 for content release assay, which collects two different time points of analysis).

The density of data needs to be increased to state these numbers as a general finding of the study.

Following reviewers' advice we increased the datapoints density. For each datapoint we now have at least 4 independent experiments. For instance, in figure 2C, that originally contained only 2 independent replicates ($n=2$) at two different timepoints for 4°C binding experiments, we now report 4 independent replicates with at least 4 different timepoints. Results and conclusion, remained unchanged. Note that an independent experiment (n) represents the mean of two technical duplicates, with each duplicate representing the "average" behavior of up to 200 000 acceptor cells, our assay being a bulk assay. In total, our data represent the results of 8 experiments that emanated from up to 1 600 000 cells, which I believe is biologically informative and meaningful.

EV isolation is time consuming and labor intensive and constitute the bottleneck within our pipeline, as the reviewer is certainly aware of. The amount of EV required for running those assays, numerous control quality experiments (immunoblots, protease protection) prevent us from performing the higher number of replicates that would be required to systematically demonstrate normal distribution and generate comprehensive statistical analysis that are, for instance, used in the drug discovery field. We believe that, although interesting, this kind of statistical analysis is outside the scope of the present study. Other comments related to statistical analysis are discussed below (minor point 3).

3. Although there is no doubt that Hsp70 is EV-associated, there are several indications in the literature that it can leave the cell via other routes. Fig. 1 D indeed shows that protease digestion of EVs reduces the Hsp70 cargo by about 25 %, suggesting that a significant part of Hsp70 in the 100.000 x g pellet is not membrane-protected and appears to be present outside EVs. The level of EV-association of Hsp70 could be further explored using density gradient centrifugation or size exclusion chromatography. Analysis of other typical cytosolic EV-cargoes, such as Alix or syntenin, would further increase the body of evidence and verify the results (or reveal that there are other EV-subtypes with distinct uptake and cargo-release kinetics, which would not be surprising). However, I see that analysis of other cargoes in the assay may be beyond the scope of the present manuscript. This is a very important point that was shared also by the reviewer 1. We used floatation assay to isolate NLucHsp70 and NLucCD63 containing EVs. Both cargoes were mostly associated with the 30% sucrose fraction (supplementary figure 1), demonstrating that NLucHsp70 is indeed associated with EVs. Importantly, when treated by proteinase K, "30% floating EVs" containing NLucHsp70 and NLucCD63 again showed 14+/-4% and 13+/- 7% of "unexpected" degradation of their cargoes, respectively. Importantly, endogenous Alix showed similar behavior. This is now reported in supplementary figure 1A, D and results are described in the core of the manuscript. Our simplest explanation is that EV isolation procedure partially damages the EV membrane integrity. Importantly, uptake of EV isolated by floatation is similar to the one reported for EVs isolated by centrifugation alone. Recombinant free Luciferase is not uptaken by acceptor cells. Altogether these new results confirm our initial data and strongly suggest that we are following the fate of cargo contained within EVs.

4. The study compares CD63 as a membrane-associated and Hsp70 as internal EV cargo. Cargo release is measured only for Hsp70 in the cytosolic fraction and ignored for CD63, which actually could be measured in the membrane fraction. In the turn of membrane fusion, CD63 is incorporated in the cellular membrane pool and will become protease sensitive (while non-fused EVs would be hidden in the endosomal compartment and protease-resistant). The relation between total membrane-associated NLuc-CD63 and protease-sensitive NLuc-CD63 would give information on the release rate of CD63. Have the authors performed such experiments?

This is a great suggestion to upgrade the present assay. Unfortunately, we tried and realized that such a nontrivial experiment requires extensive development that would delay the publication of our study. Major reasons are that 1) additional centrifugation steps introduced significant loss of material and forced us to deal with significantly lower luciferase activity, 2) even when NLucCD63 is partially digested, we cannot rule out that proteinase K has been

bulkily loaded within the intra luminal vesicles contained within the MVB. The general setup of the proposed experiment is indeed very similar to cell free assay that addressed cargo loading within exosomes (Shurtleff et al 2016, eLife).

Interestingly, a paper has been published during the preparation of our manuscript and proposed that the cytosolic tail of CD63 of uptaken EVs is accessible to the cytosol of acceptor cells, consistent with the present proposition from this reviewer (Joshi et al,2020, Acs nano). We now cite this paper in our manuscript and mention that, while consistent with our finding, this study does not formally rule out the possibility that this result might reflect bulkily loading of part of the cytosol within internalized EVs of the acceptor cell . We believe that this point also strengthens our methods to directly assess the presence of the cargo within the cytosol, that avoid the aforementioned ambiguities.

Furthermore, Fig. 3G indicates that 10 % of NLuc-C63 can be recovered from the cytosol. Is this due to membrane background in the cytosolic fraction and does this finally mean that the 30 % recovery rate should be corrected by this background value?

We now more rigorously reported the data in the core of the manuscript. For instance NLuc activity in the cytosol emanating from NLucHsp70-EV-is 27+/- 7 %, and 11+/- 8 from NLucCD63-EV. The latter is comparable to NLuc activity measured in the cytosol of donor cell expressing NLucCD63 8+/-4, the experimental background of our biochemical assay. As explained below (minor point 3), we applied on data from figure 3G Shapiro-Wilk test followed by one-way ANOVA test to assess and validate differences between the Hsp70 and CD63 related data.

Importantly, and as extensively explained in the minor comment section below, we believe that the strength of our study relies on parallel and independent tests/assays that are used to address the same point.

Concerning the EV cargo cytosolic release, EV-derived NLucHsp70 in the cytosol is reduced by either Bafilomycin treatment and IFITM-overexpression, using the same cell-based assay. In addition, these results were independently confirmed with confocal imaging. In the IFITM-related section, we even added a third independent assay. All the results, obtained independently, converge.

We understand that statistic validation for each individual set of data can be challenged, but we are confident that such questioning is counterbalanced and rigorously answered when 2 to 3 independent assays lead to the same results.

Our results evidence the cytosolic release of cargo contained within EV and we propose a rough estimation (30%) based on our best method so far. Quantification of this process has never been done before and we believe that our study establishes a first estimation that will serve as a reference until we have more accurate method.

Historically, one strength of biochemistry has been to provide quantifications and initial estimations of basic processes. This has been pivotal to further understand the mechanisms that regulate those functions. Few relevant examples: 1) plasma membrane turnover through endocytosis (roughly 50% per hour), 2) rate of autophagic degradation (1 % per hour in hepatocytes) 3) proportion of proteins following the secretory pathway (30%). We believe that our rough and yet robust estimation of EV cargo cytosolic release (30 % in our system) constitutes such a step.

5. Based on their results, can the authors exclude fusion at the plasma membrane? The requirement of acidification for content release may be a strong argument for endosomal release, but how about local pH-changes at the plasma membrane? Could these pH changes be sufficient to drive EV-fusion?

We cannot entirely rule out that some EVs used plasma membrane as a target membrane to fuse when locally, the pH has been acidified. This is now mentioned in the text. However, this seems very unlikely, for several reasons. 1) We failed to detect EV docked at the cell surface even after 2 hour incubation at 4⁰C, 2) Bafilomycin treatment increases the number of GFP-EV foci colocalizing with Rab5 endosomes but not at the cell surface (figure 4C, D

and Supplementary Figure 2B), 3) IFITMs decrease EV-content release and colocalize with EV cargo within intracellular compartments (i.e endosomes), but did not increase accumulation of EVs at the cell surface. Those results, and known delivery mechanisms used by pH-dependent viruses (VSVG, SFV, etc) strongly suggest that EV delivery seems to mainly occur within acidic endosomes

Minor points:

1. Fig. 3 G, quantification of EV-content release: The data are expressed relative to the total activity measured in the fractions. It would be informative to see how this relates to the total activity loaded to the cell. Furthermore, the data are collected from different timepoints of EV uptake (1-4 h). It would actually be interesting to see kinetics and how cytosolic NLuc-Hsp70 develops with time.

First we would like to emphasize that calculation of % of NLuc activity associated with the cytosolic fraction and absolute value in each fraction were detailed in the method section and reported in figure 3D and E for the control samples, which served to validate the method. Concerning the “development” of cytosolic NLucHSP70 overtime. The % of cytosolic delivery, remains constant overtime and represents roughly 30% of the uptaken EV fraction. This is why we used datapoints from 1 to 4 hours. Please find below (figure 2) representative example of absolute % of NLuc (from the total input) found in the cytosol, the relative % (from the uptaken fraction) at three different time-points.

2. Fig. 2E: it is not clear how the measurement was performed (please explain in more detail at least in the methods) and what the individual data points reflect (compartments within cells?). How many cells were analyzed in total? The text claims 30 % co-localization with early endosomes at 1h; where is this information reflected in the figure?

We apologized for the confusion and the difficulties to read the graph, which has been replaced. We are now showing a new graph representing the colocalization using the same method that we used in Fig 5D, that was well received by both reviewers. The method of analysis is simplified and we now measure the intensity of fluorescence of the red signal (emanating for RAb5 or Lamp I labelling) within ROI positive for GFP-EV signal. Each dot represents a ROI. We scored colocalization as positive when the red intensity within a GFP+ ROI is superior to the mean of background of red signal in cytosol plus two standard deviation.

3. Methods/statistical analysis: please reconsider the application of student’s t-test in the designated figures (number of n appears too low to claim normality of the data).

As originally written in the method section, we initially assumed normality of the data for all experiments except confocal analysis. We thought that t-test, that is often asked and used in similar studies published in top journal by well-established groups (no citation on purpose), was appropriate to support each individual set of data statistics. However, thanks to reviewer 2’s comment we now have systematically assessed normality using the Shapiro-Wilk test. Indeed, only figure 3G data passed this test. We performed one-way ANOVA test on those data, and as with the “inappropriate” t-test, statistical significance was reached.

Importantly except for Figure 3G, all the other data were confirmed using at least two independent methods, and we believe that this is the strength of our study. For instance, in some cases, such as inhibition of EV content release by IFITMs assessed by bulk cell-free assays (figure 5F, G), we could not apply rigorous statistics, due to the nature of the experiments. But this “strong” result, was confirmed with cell imaging that has been

quantified (figure 5 C, D), and the Nluc-cargo delivery cell-based assay that has been repeated at least three time (figure 5A, B).

We hope that the lack of statistics, when tests were not adequate, is counterbalanced by our combination of parallel and independent assays, which provided convergent results that led to our conclusions and interpretations.

4. Methods: please explain acronym PPI.

As originally mentioned in method section of the manuscript PPI stands for Protease/Phosphatase Inhibitor cocktail, used in our lysis buffer.

REVIEWERS' COMMENTS

Reviewer #1 (Remarks to the Author):

The authors have done an excellent job of addressing the comments and the article, which was already very good, is now even better.

Reviewer #2 (Remarks to the Author):

The authors satisfyingly addressed the main issues and provided further critical controls. The manuscript has been improved significantly and I do not have any further comments.

Overall, the study represents a significant technical advance, which is important for the development of this dynamic field and of great value to a broad readership.

Eva-Maria Krämer-Albers